# Self-Assessed Experience of Emotional Involvement in Sensory Analysis Performed in Virtual Reality

**DOI:** 10.3390/foods13030375

**Published:** 2024-01-24

**Authors:** Abdul Hannan Bin Zulkarnain, Xu Cao, Zoltán Kókai, Attila Gere

**Affiliations:** Institute of Food Science and Technology, Hungarian University of Agriculture and Life Sciences, Villányi út. 29-31, H-1118 Budapest, Hungary; zulkarnain.abdul.hannan.bin@phd.uni-mate.hu (A.H.B.Z.); cao.xu@phd.uni-mate.hu (X.C.); kokai.zoltan@uni-mate.hu (Z.K.)

**Keywords:** food sensory, multiple factor analysis, PANAS, emotion, VR

## Abstract

Virtual reality (VR) technology has gained significant attention in various fields, including education for health professionals, sensory science, psychology, and consumer research. The first aim of the paper is to explore the self-assessed experience of emotional involvement in sensory analysis performed in VR. The Positive and Negative Affect Schedule (PANAS) is a widely used self-report measure that assesses positive and negative affective states. VR sensory analysis involves the use of immersive, interactive, and multi-sensory environments to evaluate sensory perception and emotional responses. By synthesizing relevant literature, this paper provides insights into the impact of VR on affective states, the effectiveness of VR in eliciting emotions, and the potential applications of the PANAS in VR sensory analysis. Furthermore, the second aim of the paper is to uncover the effect of VR sensory evaluation on the participant’s emotional states, as it has a significant effect on their evaluations. The results suggest an increase in the sum of positive effects and a decrease in the negative ones. Although these results are promising, the relationship between the PANAS and VR sensory analysis is still underexplored, with limited research investigating the specific effects of VR on affective states measured using the PANAS. Further research is needed to better understand the potential of the PANAS in assessing emotional responses in VR environments and its implications for sensory analysis.

## 1. Introduction

Virtual reality (VR) technology has revolutionized various fields, including education, healthcare, entertainment, and consumer research [1]. VR provides users with immersive and interactive experiences in computer-generated environments, stimulating multiple sensory modalities such as vision, hearing, and touch [2]. This technology has been widely used for educational applications in health professions to enhance learning outcomes and improve healthcare professionals’ knowledge, cognitive skills, attitudes, and satisfaction [3]. In sensory science, VR has emerged as a promising tool for evaluating sensory perception and emotional responses to various stimuli, including food products [1]. Understanding the relationship between affective states and VR sensory analysis is crucial for leveraging the potential of VR in enhancing sensory evaluation and consumer research.

### 1.1. Positive and Negative Affect Schedule (PANAS)

VR technology has provided a new way to understand sensory perception. The immersive and interactive experiences extend beyond entertainment and gaming to various fields, including education, marketing, and healthcare [2]. Regarding the context of sensory analysis and emotional perception, the integration of VR presents numerous benefits for exploring the affective states and sensory responses of individuals in simulated environments. This section aims to provide an overview of the Positive and Negative Affect Schedule (PANAS), the impact of VR on affective states, and the effectiveness of VR in eliciting emotions for understanding emotional responses and sensory perception in VR environments.

The PANAS questionnaire is a widely used self-report questionnaire designed to measure positive and negative affect in many research fields. It consists of a list of adjectives that describe different feelings and emotions, and respondents are asked to indicate the extent to which they have experienced each emotion during a specific time frame [4]. The PANAS has been adapted and utilized in various cultural and linguistic environments to measure affective states in different populations [5,6]. As for sensory analysis, the PANAS is usually applied to measure the affective states of individuals involved in sensory evaluation processes. Moreover, sensory analysis allows for the evaluation and description of the sensory attributes of products or experiences, which involves systematically assessing sensory characteristics such as taste, smell, texture, and appearance using human sensory perceptions [7]. The affective states of individuals conducting sensory descriptive analysis can influence their perception and evaluation of food products [8].

Additionally, sensory analyses of food products, such as yogurt, and defective fruits and vegetables, often involve subjective assessments that might be again influenced by the affective states of the evaluators [9]. When conducting VR-based sensory analysis, the PANAS can be a robust instrument for quantifying the emotional states of individuals immersed in virtual environments, thereby contributing to a deeper understanding of the interplay between affective states and sensory perception within VR settings. Therefore, applying the PANAS in sensory analysis can provide insights into how affective states impact sensory perception and evaluation, contributing to a more comprehensive understanding of the sensory experience of food [10,11,12,13]. 

The PANAS method is very popular and well-established, demonstrating positive and negative aspects. Versatility, psychometric properties, and cross-cultural validity are the positive aspects. Many studies employed the PANAS method and validated it in different languages and cultural contexts, such as Spanish, Chinese, and Polish, indicating its versatility and applicability across diverse populations [14]. Research has demonstrated the psychometric properties of the PANAS, including good internal consistency and reliability [4,15]. However, there are negative aspects as well. There has been an ongoing debate regarding the factor structure of the PANAS, indicating varying factor structures and measurement invariance across different populations and cultural backgrounds [16,17]. Besides, there exists a response bias of the PANAS. As a self-report measure, the PANAS is susceptible to response biases and individual differences in emotional expression, which may impact the accuracy of the reported affective states. Despite these considerations, the PANAS’s consistent application and validation underscore its significance as a self-report measure in assessing affective states, particularly in the context of VR sensory analysis.

### 1.2. Impact of VR on Affective States

In recent years, researchers have increasingly focused on integrating VR technology into the domain of sensory analysis studies. Due to its excellent spatial environment simulation and the low cost of rapidly changing experimental environments, it has a high potential to influence affective states and emotional experiences. VR has significantly influenced affective states, eliciting emotional responses and sensory perceptions that closely mirror real-world experiences [18,19,20]. Additionally, exposure to VR has been found to increase positive and decrease negative affective states, making it a potential tool for mood improvement [21]. However, it is essential to note that the impact of VR on affective states is not limited to positive experiences, as negative emotional consequences of VR gameplay have also been discussed. Ślósarz et al. [22] found that VR significantly modifies learners’ emotions, reinforcing positive emotional states, and decreasing negative emotional states. Furthermore, the potential of VR to influence affective states has also been explored in the context of exercise games, pediatric pain management, and interventions for people living with dementia and mild cognitive impairment, underscoring its diverse applications in promoting emotional well-being and enhancing psychological health [23,24,25].

The influence of VR on affective states extends to various domains, including tourism, education, language learning, and historical education [26,27,28]. Additionally, it was found that using VR in history education can promote enjoyment and influence morality, further emphasizing its impact on affective states [29]. For instance, the Palace Museum (Beijing, China) has created an interactive experience program based on VR technology, featuring the famous painting “Along the River During the Qingming Festival”. This project enables participants to deeply reconstruct the scenes depicted in the painting, immersing them in the detailed portrayal of the thriving Song Dynasty era.

It is worth noting that the impact of VR on affective states is not uniform across all experiences [30]. On the one hand, the effectiveness of VR in enhancing store attractiveness has been highlighted, indicating its potential to influence emotional responses in specific environments. On the other hand, the impact of sensory cues offered by VR on pro-environmental behavioral intention has been reviewed, suggesting a broader influence of VR on emotions and behavior. In summary, the impact of VR on affective states is multifaceted, with studies demonstrating its potential to elicit emotional responses, influence mood states, and modify learners’ emotions across various domains. While VR has shown promise in enhancing positive affective states and immersive experiences, it is essential to consider its potential negative emotional consequences and the contextual factors that may modulate its impact on affective states.

### 1.3. Effectiveness of VR in Eliciting Emotions

VR technology can potentially influence participants’ emotional states and holds promise for emotion recognition. Assisted by the Large Language Models, it is possible to anticipate and guide emotions in advance. Liu et al. [31] explored the effects of viewing a 360-degree video on emotional well-being among elderly people and college students in immersive VR conditions. Their study shows that VR has significant potential in adjusting emotional states. Similarly, Marín-Morales et al. [32] highlighted the potential of VR in emotion recognition, particularly in immersive environments. Their research underlined the synergy between VR, implicit measurements, and machine-learning techniques and their impact on various research areas. 

Humans perceive the world through sensory cognition, with vision being one of the primary cognitive pathways. VR technology has created an immersive visual environment and can recognize emotion through immersive experiences [33]. These studies underscore the significance of VR in eliciting emotional responses, providing a foundation for studying emotional and sensory perception in VR environments. Furthermore, VR has huge potential in promoting learning experiences in contexts such as medical education. The application of VR in providing immersive experiences for students demonstrated its potential to influence sensory perceptions and emotional engagement [34]. Similarly, Torrico et al. [35] investigated the effects of emotional responses and VR on the wine-tasting experience, and they found that VR environments influence sensory experiences and emotional responses, which indicates the potential of VR to modulate perceptions and affective states in sensory analysis. 

This finding underscores the potential of VR to modulate perceptions and affective states in sensory analysis. In synthesis, this introduction provides valuable insights into VR’s multifaceted impact and effectiveness in eliciting emotions. It highlights its diverse applications, particularly in the nuanced field of sensory analysis.

### 1.4. Positive and Negative Affect Schedule (PANAS) Questionnaire

The PANAS is a widely used questionnaire designed to measure the two primary dimensions of mood: positive affect (PA) and negative affect (NA). Developed by Watson, Clark, and Tellegen in 1988 [36], the PANAS has been utilized in numerous research studies across various disciplines, including psychology, psychiatry, and health sciences. The questionnaire consists of two 10-item scales, one for measuring positive affect and the other for measuring negative affect. Respondents are asked to rate the extent to which they have experienced each emotion over a specific period, typically ranging from “very slightly or not at all” to “extremely”. The PANAS has been translated into multiple languages and validated in diverse cultural contexts, making it a valuable tool for cross-cultural research on affect and mood.

The Positive Affect Scale (PAS) of the PANAS includes items such as “interested”, “excited”, and “enthusiastic”, while the Negative Affect Scale (NAS) includes items such as “distressed”, “irritable”, and “scared”. The PANAS has demonstrated good internal consistency, test–retest reliability, and convergent and discriminant validity. It has also been used in clinical settings to assess changes in affective states before and after interventions and in epidemiological studies to investigate the relationship between affect and various health outcomes.

One of the key strengths of the PANAS is its brevity, which allows for the efficient assessment of affective states without imposing a significant burden on respondents. Additionally, the PANAS is sensitive to experimental manipulations of mood, making it a valuable tool for studying the impact of interventions or environmental factors on affective states. However, it is essential to note that the PANAS primarily measures the intensity of affective experiences and may need to capture the full complexity of emotional processes. Researchers and clinicians should, therefore, consider complementing the PANAS with other measures of emotion regulation, emotional granularity, or specific emotional states when aiming to provide a comprehensive assessment of affective functioning.

The PANAS questionnaire is a valuable instrument for assessing positive and negative affective states and has been widely used in research and clinical settings. Its brevity, reliability, and cross-cultural applicability make it versatile for studying mood and emotion across diverse populations. Researchers and practitioners should be mindful of the limitations of the PANAS and consider integrating it with other measures to obtain a more comprehensive understanding of affective experiences.

### 1.5. Applications of Positive and Negative Affect Schedule (PANAS) in VR Sensory Analysis

The use of VR in sensory analysis offers several advantages over traditional methods. Kong et al. [37] conducted a preliminary study on the sensory perception of chocolate products in immersive VR environments. They found that VR provided better engagement and ecological validity than traditional sensory booths. Crofton et al. [38] highlighted the potential application of VR technology in creating immersive contextual settings for hedonic testing in sensory science. The studies suggest that VR can enhance sensory evaluation by providing a more realistic and immersive experience [39]. However, the specific application of the PANAS in VR sensory analysis still needs to be explored. The PANAS could be used to assess emotional responses in VR environments and provide valuable insights into participants’ affective states during sensory evaluation. Further research is needed to investigate the feasibility and effectiveness of using the PANAS in VR sensory analysis.

In the context of sensory analysis, the PANAS has been used to evaluate participants’ emotional responses to different sensory stimuli. For example, a study by Mielmann et al. [40] investigated the impact of mood, familiarity, acceptability, sensory characteristics, and attitude on consumers’ emotional responses to chocolates. The researchers used the PANAS to assess participants’ positive and negative affective states after consuming different chocolates. The results showed that various factors, including sensory characteristics and individual differences in mood and attitude, influenced emotional responses.

Furthermore, the PANAS has been used in food science to investigate the relationship between sensory attributes and emotional responses. Pramudya and Seo [41] explored the influences of product temperature on emotional reactions to and sensory characteristics of coffee and green tea beverages. They used the PANAS to assess participants’ emotional responses and found that product temperature significantly influenced sensory attributes and emotional responses.

In addition to food and beverages, the PANAS has been used in other sensory-related studies. For example, a meta-analysis by Satpute et al. [42] investigated the involvement of sensory regions in affective experience. The researchers found that activity in early sensory areas may provide information pertinent to other sensory modalities. This suggests that the early sensory cortex may be an association cortex for other stimulus modalities. Although the PANAS was not directly used in this study, it provides insights into the relationship between sensory processing and affective experience.

Overall, the PANAS is a valuable tool in sensory analysis for assessing individuals’ emotional responses to sensory stimuli. It allows researchers to examine the relationship between sensory attributes and emotional states, providing insights into the affective experience of individuals in various contexts.

The first aim of this study is to review the use of the Positive Affect Negative Affect Schedule (PANAS) in VR and sensory analysis, thus setting the ground for the use of the PANAS in VR sensory evaluations. The focus is on establishing a foundation for the subsequent use of the PANAS in VR sensory assessment. Since this study has a specific aim, the PANAS is incorporated into a VR sensory evaluation to systematically assess the impact of VR sensory experiences on participants’ emotional states. Considering the widespread utilization of VR applications in the food industry, particularly in enhancing interactions between customers, food products, and context, this research addresses a critical gap by investigating emotional measures in VR. Despite numerous sensory tests conducted in VR, a greater understanding of emotions during VR usage is needed. This study aims to contribute to the innovative integration of emotional measures, specifically using the PANAS, in VR sensory evaluations, thereby advancing our comprehension of emotional involvement in this dynamic virtual environment.

## 2. Materials and Methods

### 2.1. Participants

The participants were Hungarian University of Agriculture and Life Sciences (MATE) students. Based on Table 1, the sample was composed of forty-two (42) participants, 62% of which were female (with a mean age of 25.50 ± 2.97) and 38% of which were male (with a mean age of 25.19 ± 3.10). Participants reported needing more experience with VR before.

Participants gave verbal consent via the statement, “*I am aware that my responses are confidential, and I agree to participate in this experiment*”, where an affirmative reply was required to participate in the experiment. They could withdraw from the experiment at any time without giving a reason. The products tested were purchased commercially and, therefore, safe for consumption. Ethical approval was obtained from the MATE internal ethics committee (approval number: MATE-BC/947-1/2023).

### 2.2. Experimental Environment and Measurement Setup

The experimental environment was divided into two parts: a sensory booth in the sensory laboratory and an empty classroom at the Hungarian University of Agriculture and Life Sciences (MATE) dedicated to the VR experiment. The virtual sensory laboratory was virtualized and designed using Unity version 2022.3.10f1 (Unity Technologies, Unity Software Inc., San Francisco, California, USA), with head-mounted displays (HMD), HTC VIVE Pro Eye (HTC Corporation, Xindian, New Taipei, Taiwan), and a hand and finger motion sensor Leap Motion Controller (Leap Motion, Inc., San Francisco, California, USA). One student assistant was recruited to help set up the system and instruct the participants about the rules to follow during the experiment (or on the experiment’s rules).

### 2.3. Software Development and Virtual Sensory Booth

The software was designed using Unity version 2022.3.10f1 (Unity Technologies, Unity Software Inc., San Francisco, CA, USA). Figure 1 shows a replica of the VR sensory booth. The VR sensory booth was designed to be as identical to a standard sensory booth as possible. Based on the ISO 8589: 2007 standard [43], a well-established sensory laboratory must use white (or light grey) colors, good natural lighting (6500 K), and the air must be well-ventilated. The virtual sensory booths were 1 m × 1 m × 2.5 m (w × d × h). They were completed with a computer, a monitor, a chair, and samples indicated with a three (3) digit randomized code and a glass of water.

### 2.4. Procedure

The flow of the virtual sensory testing experiment is shown in Figure 2. Participants attended during their booked appointment and gave their consent to participate in the experiment.

Before beginning, participants were briefed about the study’s expectations and aims again. First, it was necessary for participants to fill out online forms with demographic questions using a tablet. The participant was placed on an empty table with lemonade samples with three different sugar concentrations (10%, 20%, and 30% sweetness). Next, they were assisted by a laboratory assistant (LA) in putting on head-mounted displays (HMDs). Once the HMDs were on, the VR sensory booth started with the same layout as a traditional sensory booth (Figure 3). The samples were presented virtually with three randomized digits and were placed on a red marker in a virtual cup as a palate cleanser. In the VR environment, participants could quickly locate the samples on the marker, showing their position (Figure 3). With the assistance of the LA, participants answered orally about their preferences (sweet, sour, and overall liking) on a nine-point hedonic scale that ranged from one (dislike highly) to nine (like extremely). The LA then recorded the ranking in the online questionnaire on the tablet. After the sensory testing, the participants’ HMDs were removed, and they were asked to complete the post-VR questionnaire on the tablet. The results were generated automatically online. The participant was given a candy as a gift for participating.

### 2.5. Measures

The participants were required to complete a series of questionnaires and sensory testing as part of the experiment.

The pre-experiment questionnaire consisted of the PANAS questionnaire, and demographic information including sex, gender, age, and nationality was collected. VR experience-related information (e.g., familiarity) was also collected for a separate investigation.

During the sensory testing, participants expressed their evaluations verbally, and the student assistant recorded the answers.

For the post-experiment questionnaire, participants completed the PANAS questionnaire, post-VR questionnaire, and comment section.

#### Positive and Negative Affect Schedule (PANAS) Questionnaire

The Positive and Negative Affect Schedule (PANAS) is a widely used self-report questionnaire designed to measure the two broad dimensions of mood: positive affect (PA) and negative affect (NA) [36]. Table 2 shows the questionnaire, consisting of two separate 10-item scales, one for PA and one for NA. The positive emotions are as follows: interested, excited, strong, enthusiastic, proud, alert, inspired, determined, attentive, active, and distressed. Meanwhile, the negative emotions are as follows: upset, guilty, scared, hostile, irritable, ashamed, nervous, jittery, and afraid.

Using a Likert-type scale, participants were asked to rate the extent to which they experienced each emotion or feeling during the VR sensory test. The scale ranged from 1 (very slightly or not at all) to 5 (extremely).

Scoring:The Positive Affect Score was calculated as follows: add the scores on PANAS items 1, 3, 5, 9, 10, 12, 14, 16, 17, and 19. Scores can range from 10–50, with higher scores representing higher levels of positive affect.The Negative Affect Score was calculated as follows: add the scores on PANAS items 2, 4, 6, 7, 8, 11, 13, 15, 18, and 20. Scores can range from 10–50, with lower scores representing lower levels of negative affect.

### 2.6. Data Analysis

The findings were statistically interpreted and displayed in tabular and graph form, with the mean or average value, minimum, maximum, and standard deviation. A multivariate analysis approach was applied to the result of the PANAS score using (*p* < 0.05) XLSTAT (Addinsoft, New York, NY, USA). A Multiple Correspondence Analysis (MCA) was performed on the PANAS score and emotions. Figures were prepared using XLSTAT (Addinsoft, New York, NY, USA).

## 3. Results and Discussion

### 3.1. Positive and Negative Affect Schedule (PANAS)

#### 3.1.1. Overall Positive and Negative Affect Schedule (PANAS) Scores

In Figure 4, the PANAS score indicates that positive emotions before the experiment had a mean score of 32.79 ± 10.19, which increased to 35.33 ± 9.12 after the experiment. However, there was no significant difference (*p*-value = 0.115) in the participants’ emotional states before and after the experiment. On the other hand, the negative emotions before the experiment had a mean score of 15.31 ± 7.34, which decreased to 12.52 ± 3.88 after the experiment. This indicates a statistically significant difference in the participants’ emotional states before and after the experiment (*p*-value = 0.016).

Previous studies showed that VR was associated with increases in positive emotions and decreases in negative emotions. Yeo et al. [20] found that computer-generated VR was linked to significantly more significant improvements in positive affect compared to other media, mediated by a greater experienced presence, and increases in connectedness with nature. Similarly, Browning et al. [44] observed that positive affect remained constant in the virtual condition while negative affect decreased. Furthermore, Ślósarz et al. [22] reported a significant increase in positive emotions following VR intervention, compared to negative emotions during the post test. These findings collectively support the notion that VR experiments can lead to an increase in positive emotions and a decrease in negative emotions.

Moreover, Pavic et al. [45] highlighted encouraging results regarding the effectiveness of VR in fostering positive emotions. Additionally, Basbasse et al. [46] suggested that stronger experiences of emotions, particularly fear, in VR tasks are associated with higher levels of asymmetry for negative emotions. This indicates that VR can elicit intense emotional responses, potentially leading to a decrease in negative emotions. Furthermore, Liu et al. [31] found that using VR headsets significantly increased self-efficacy, increased positive emotions, and decreased negative emotions in patients with fibromyalgia.

However, it is essential to note that VR experiences can also have potential negative emotional consequences. Lavoie et al. [47] revealed that intensified negative emotions resulting from VR were significantly correlated with negative rumination. Similarly, Frentzel-Beyme and Krämer [27] discussed how emotionally charged historical VR experiences might decrease critical, cognitive reflection and lead to strong emotional reactions. Therefore, while VR experiments have the potential to increase positive emotions and decrease negative emotions, they may also have adverse emotional effects.

#### 3.1.2. Individual PANAS Items

Table 3 shows the average of emotions before and after the experiment with a *p*-value showing the results of two-sample *t*-tests.

Several emotions had significance before and after the experiment. The positive emotions that had significant differences and increased after the experiment were “interested”, “excited”, “proud”, and “inspired”. The Negative emotions that decreased were “guilty”, “ashamed”, “nervous”, and “afraid” (Table 3).

#### 3.1.3. Participants Positive and Negative Affect Schedule (PANAS) Score

Figure 5 shows the individual PANAS scores. A total of 61.90% of the participants experienced increased positive emotions, while 38.10% experienced decreased positive emotions. Meanwhile, 57.14% of participants experienced decreased negative emotions, and 7.14% experienced increased negative emotions. Individuals with significant positive emotion differences are P2, P33, P34, P37, P38, P39, and P40. At the same time, highly significant differences in negative emotions were found for P7, P22, P29, P31, and P32.

This reinforces the discussion regarding the interplay between VR sensory evaluation and participants’ emotional states. The observed rise in positive emotions aligns with the immersive nature of VR experiences, suggesting their potential to evoke positive effects. Simultaneously, decreasing negative emotions implies a positive emotional impact associated with engaging in VR sensory evaluations. These findings contribute to a comprehensive understanding of how VR environments influence and enhance emotional states, highlighting the potential for positive emotional effects and reducing negative emotional responses within the sensory analysis.

The use of VR has been shown to have a significant impact on individuals’ emotional states. Several studies have demonstrated that VR interventions can lead to an increase in positive emotions and a decrease in negative emotions. For instance, Browning et al. [44] found that adverse effects decreased after exposure to 360-degree nature videos in VR. Similarly, Ślósarz et al. [22] observed a significant increase in positive emotions following a VR intervention, compared to the intensity of negative emotions. Moreover, Pavic et al. [45] highlighted the effectiveness of VR in inducing positive emotions across various settings and adult lifespan. Lavoie et al. [47] also reported significantly reduced negative emotions in individuals exposed to a VR-based restorative environment. They suggested that VR tasks evoked more realistic fears and could lead to intensified negative emotions.

However, it is essential to note that the impact of VR on emotions is not universally positive. It has been found that negative emotions intensified by VR were correlated with negative rumination; for example, Basbasse et al. [46] indicated potential negative emotional consequences of VR experiences. Furthermore, Li et al. [48] highlighted that the negative effects of immersive VR were associated with a reduction in felt pleasantness, indicating potential negative emotional outcomes.

### 3.2. Multiple Factor Analysis (MFA) on Positive and Negative Affect Schedule (PANAS)

In the experiment, the data in Figure 6 illustrate the MFA of both positive and negative emotions before and after the experiment. This can be validated further by Figure 4, which shows that the positive emotions increase while negative emotions decrease after the experiment.

Furthermore, according to Figure 7, VR has the potential to significantly impact emotional experiences, especially in enhancing positive emotions and reducing negative emotions. The study suggests that VR can successfully induce positive emotional states, which ensures that no bias is introduced in the sensory test due to any changes in the participant’s emotional state while working in a VR environment.

The role of VR in eliciting positive emotions was also explored in various contexts. Wang et al. [49] demonstrated the role of emotional responses in VR exhibitions, where participants reported feeling pleasure and satisfaction, indicating the potential of VR environments to evoke positive emotions. Additionally, Mahmud et al. [50] found that exposure to relaxing virtual environments induced positive emotions and reduced negative emotions, highlighting the potential therapeutic effects of VR in promoting positive emotional experiences.

Furthermore, the impact of VR on emotional empathy was investigated, with reports indicating that VR increases emotional empathy. In particular, Martingano et al. [51] suggested the potential of VR to enhance positive emotional connections. Additionally, it was shown that using VR in mindfulness skills training exercises reduces negative emotions and increases positive emotions in individuals. Gomez et al. [52] indicated VR’s potential in promoting positive emotional well-being.

The impact of VR on emotions is multifaceted, with studies demonstrating both positive and negative emotional outcomes. While VR can reduce negative emotions, as shown in Figure 8, it can also intensify negative emotions and harmful self-related thoughts. Meanwhile, the specific VR context and content influence emotional experiences in VR environments.

Chirico et al. [53] have highlighted that VR has the potential to elicit both positive and negative emotions, indicating that emotional experiences in VR environments are of a dual nature. Meanwhile, Wang et al. [49] have emphasized that the emotional impact of VR is influenced by the specific VR context, as demonstrated by the processing of balanced words. Furthermore, Ślósarz et al. [22] have observed an increase in the intensity of positive emotions following VR intervention, compared to the intensity of negative emotions during the post test, indicating a potential positive influence of VR on emotions. Similarly, Pallavicini and Pepe [54] have found that VR content, including VR video games, can effectively induce positive emotions and decrease negative emotions and anxiety in individuals, further supporting the potential positive impact of VR on emotions.

However, Lemmens et al. [55] demonstrated that commercial VR games can affect feelings of presence and players’ physiological and emotional states, indicating the potential for negative emotional effects. Stallmann et al. [56] also expected participants to react negatively to being excluded in immersive VR, suggesting that VR experiences can elicit negative emotions such as ostracism and exclusion.

### 3.3. Post-VR Questionnaire

The post-VR questionnaire investigates the acceptability of a virtual sensory booth (SB). It comprises five questions, as shown in Figure 9, addressing the level of immersion, the quality of graphics, the ability to pick up and place items in the virtual environment, the overall quality of the VR technology, and the overall experience with VR. Participants provide ratings for the virtual SB by selecting a value on a parameter scale between 1 (very low/very difficult/negative) and 9 (very high/very easy/positive), with higher values indicating a more favorable experience (Likert Scale). 

All the scores obtained from the post-VR questionnaire are above 7, indicating that the participants received the virtual SB well. The highest average score was given to ‘Overall experience with VR’ (8.17 ± 1.21), whereas the lowest was to ‘Pick up and/or place items in the virtual environment’ (7.10 ± 1.95). The scores for the other questions, in descending order, are as follows: ‘The quality of the VR technology overall’ (7.90 ± 1.14), ‘The level of immersion’ (7.36 ± 1.69), and ‘The quality of the graphics’ (7.26 ± 1.47). This suggests that the participants found the VR experience to be immersive.

Several studies have been conducted to explore the influence of virtual reality (VR) on emotional experiences. One such study reviewed previous research on VR, focusing on fear cues, emotions, and presence. They aimed to identify the most critical aspects of emotional experiences in VR and their interrelationships [57].

## 4. Conclusions

In conclusion, it was found that the PANAS can be effectively used in VR sensory analysis. It has been observed that VR impacts a person’s emotional state, as some studies have reported positive effects on positive affect, while others have shown no significant impact on negative affect. VR has also been proven to enhance sensory evaluation by providing a more interactive and realistic experience. However, further research is required to understand the specific effects of VR on affective states measured using the PANAS and its potential applications in VR sensory analysis. Integrating the PANAS and VR sensory analysis can provide valuable insights into emotional responses in immersive environments and contribute to the advancement of sensory science and consumer research. It is imperative to conduct further research to explore the relationship between the PANAS and VR sensory analysis comprehensively. This will aid in understanding emotional responses in virtual environments and the implications for sensory assessment.

## Figures and Tables

**Figure 1 foods-13-00375-f001:**
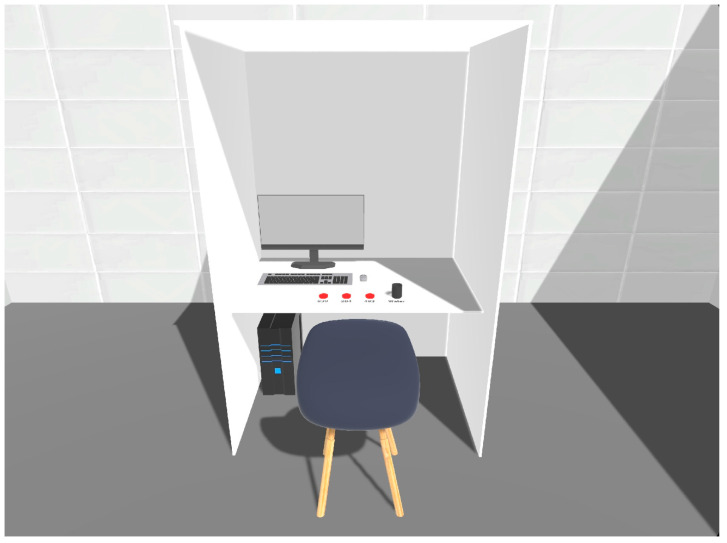
Virtual replica of a sensory booth.

**Figure 2 foods-13-00375-f002:**
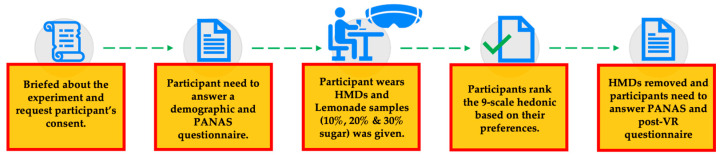
Flow of the experiment and sensory testing for participants.

**Figure 3 foods-13-00375-f003:**
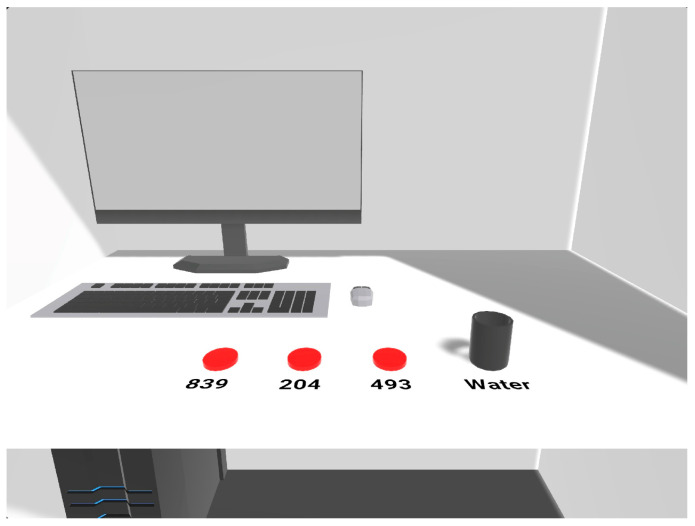
The participants’ point of view inside the virtual sensory booth.

**Figure 4 foods-13-00375-f004:**
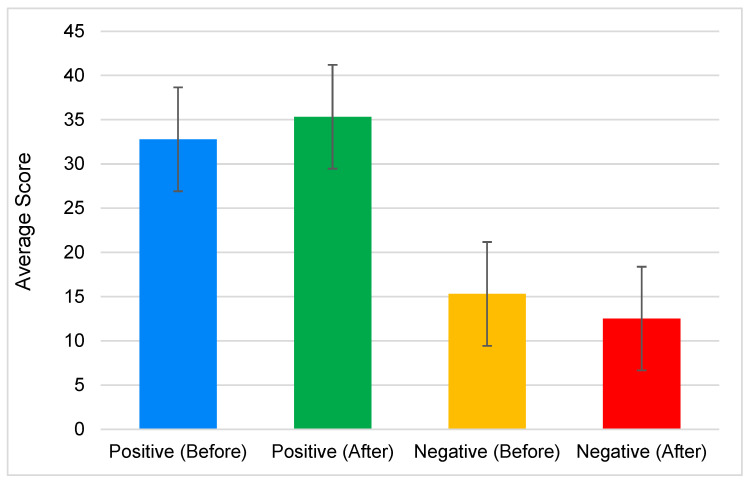
Average Positive and Negative Affect Schedule (PANAS) score before and after the experiment.

**Figure 5 foods-13-00375-f005:**
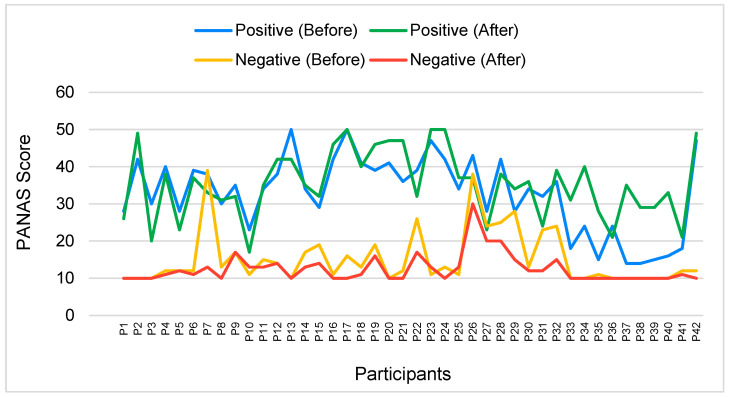
Average Positive and Negative Affect Schedule score before and after the experiment.

**Figure 6 foods-13-00375-f006:**
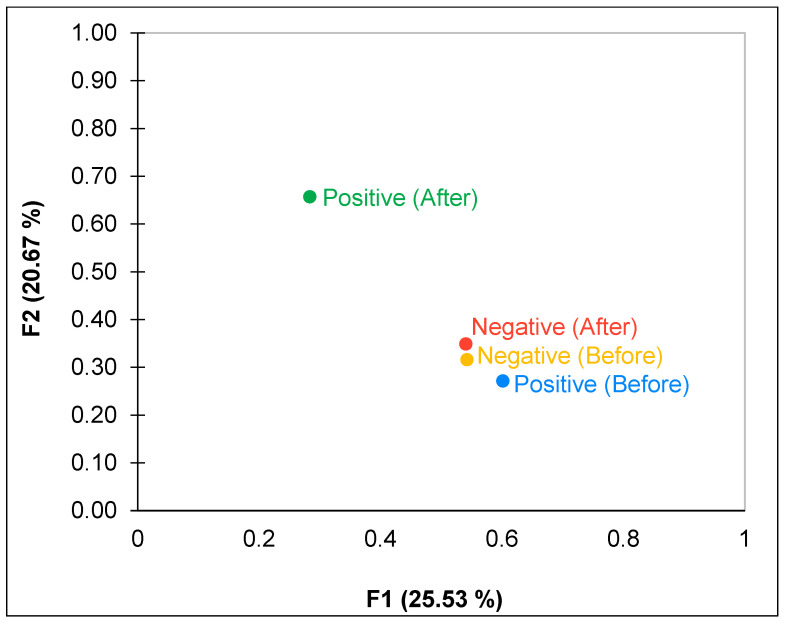
Multiple factor analysis on Positive and Negative Affect Schedule (PANAS) before and after the experiment.

**Figure 7 foods-13-00375-f007:**
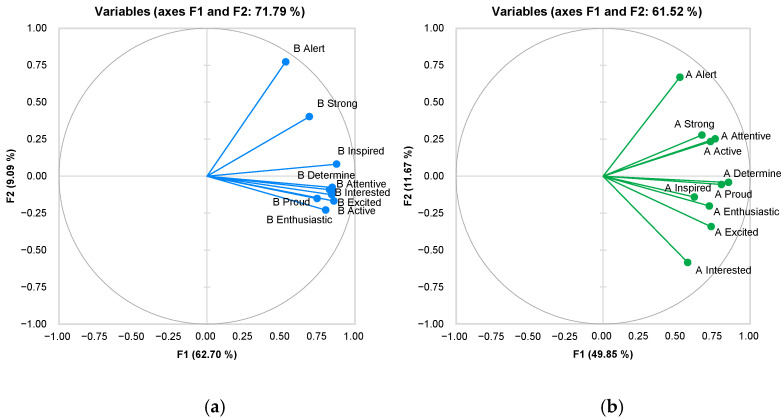
Comparison of positive emotion variables before (**a**) and after (**b**) the experiment.

**Figure 8 foods-13-00375-f008:**
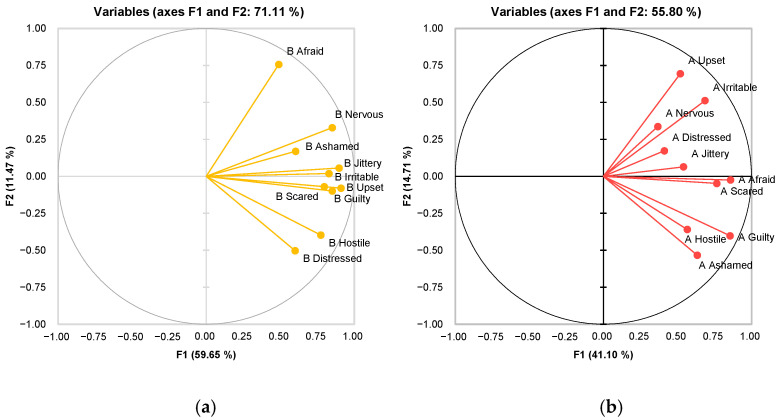
Comparison of negative emotions variables before (**a**) and after (**b**) the experiment.

**Figure 9 foods-13-00375-f009:**
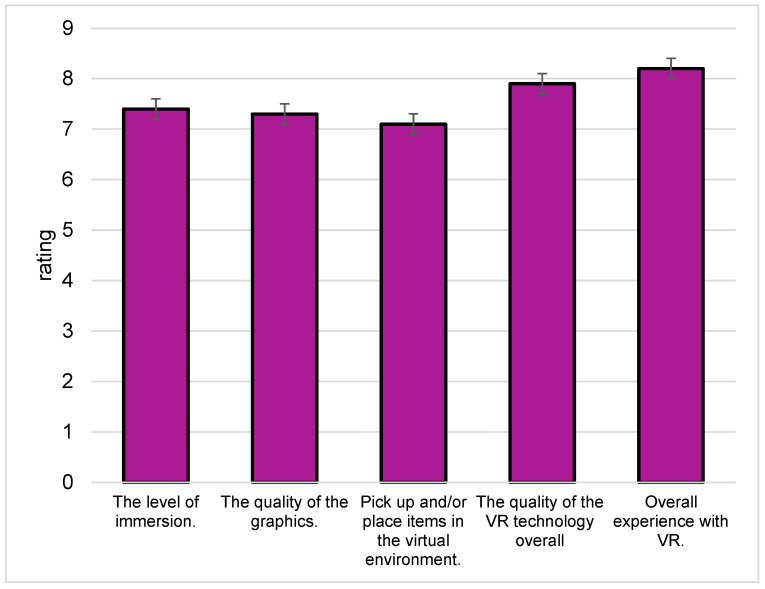
Average score of post-virtual reality questionnaire.

**Table 1 foods-13-00375-t001:** Participants’ gender, age, and virtual reality experience.

Gender	Number of Participants (*n*)	Percentage (%)	Age			VR Experience	
			Mean ± SD	Min	Max	Yes	No
Male	16	38	25.19 ± 3.10	21	33	5	11
Female	26	62	25.50 ± 2.97	21	32	10	16
Total	42	100	25.31 ± 2.98	21	33	15	27

**Table 2 foods-13-00375-t002:** Positive and Negative Affect Schedule (PANAS) questionnaire.

Emotions		Very Slightly or Not at All	A Little	Moderately	Quite a Bit	Extremely
PANAS 1	Interested	◻	◻	◻	◻	◻
PANAS 2	Distressed	◻	◻	◻	◻	◻
PANAS 3	Excited	◻	◻	◻	◻	◻
PANAS 4	Upset	◻	◻	◻	◻	◻
PANAS 5	Strong	◻	◻	◻	◻	◻
PANAS 6	Guilty	◻	◻	◻	◻	◻
PANAS 7	Scared	◻	◻	◻	◻	◻
PANAS 8	Hostile	◻	◻	◻	◻	◻
PANAS 9	Enthusiastic	◻	◻	◻	◻	◻
PANAS 10	Proud	◻	◻	◻	◻	◻
PANAS 11	Irritable	◻	◻	◻	◻	◻
PANAS 12	Alert	◻	◻	◻	◻	◻
PANAS 13	Ashamed	◻	◻	◻	◻	◻
PANAS 14	Inspired	◻	◻	◻	◻	◻
PANAS 15	Nervous	◻	◻	◻	◻	◻
PANAS 16	Determine	◻	◻	◻	◻	◻
PANAS 17	Attentive	◻	◻	◻	◻	◻
PANAS 18	Jittery	◻	◻	◻	◻	◻
PANAS 19	Active	◻	◻	◻	◻	◻
PANAS 20	Afraid	◻	◻	◻	◻	◻

**Table 3 foods-13-00375-t003:** Mean of Positive and Negative Affect Schedule questionnaire scores.

Emotions		Before	After	*p*-Value	Emotion Increased (↑) or Decreased (↓)
		Mean ± SD	Mean ± SD		
Positive	Interested	3.95 ± 1.13	4.45 ± 0.67	0.008	↑
	Excited	3.50 ± 1.27	4.00 ± 1.13	0.030	↑
	Strong	3.21 ± 1.35	3.33 ± 1.44	0.349	
	Enthusiastic	3.38 ± 1.19	3.64 ± 1.27	0.166	
	Proud	2.93 ± 1.47	3.67 ± 1.43	0.011	↑
	Alert	2.64 ± 1.39	2.21 ± 1.57	0.095	
	Inspired	3.33 ± 1.26	4.00 ± 1.10	0.006	↑
	Determine	3.07 ± 1.30	3.17 ± 1.50	0.378	
	Attentive	3.21 ± 1.42	3.07 ± 1.63	0.335	
	Active	3.55 ± 1.23	3.79 ± 1.12	0.178	
Negative	Distressed	1.88 ± 1.15	1.74 ± 1.21	0.291	
	Upset	1.48 ± 1.09	1.24 ± 0.79	0.127	
	Guilty	1.48 ± 0.99	1.10 ± 0.43	0.013	↓
	Scared	1.29 ± 0.77	1.14 ± 0.52	0.162	
	Hostile	1.50 ± 0.99	1.33 ± 0.72	0.191	
	Irritable	1.62 ± 1.13	1.33 ± 0.90	0.101	
	Ashamed	1.29 ± 0.64	1.10 ± 0.37	0.049	↓
	Nervous	1.95 ± 1.19	1.17 ± 0.44	<0.001	↓
	Jittery	1.67 ± 1.00	1.36 ± 0.82	0.063	
	Afraid	1.17 ± 0.38	1.02 ± 0.15	0.013	↓

Darker shade represents a significant increase in positive emotions, while lighter shade represents a considerable decrease in negative emotions.

## Data Availability

The data presented in this study are available on request from the corresponding author. The data are not publicly available due to the General Data Protection Regulation (GDPR) on the participant’s privacy and confidentiality.

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
