# Peer review of "Self-Assessed Experience of Emotional Involvement in Sensory Analysis Performed in Virtual Reality"

_foods, 2024, doi:10.3390/foods13030375_

Round 1

Reviewer 1 Report

Comments and Suggestions for Authors

1.      Clarified the aims and objectives of the paper regarding the exploration of emotional involvement in Sensory Analysis conducted in virtual reality (VR).

2.      Emphasized the utilization of PANAS (Positive and Negative Affective Schedule) as a self-report measure to assess affective states in VR sensory analysis.

3.      Highlighted the synthesis of existing literature to provide insights into the impact and effectiveness of VR on eliciting emotions and its potential applications in sensory analysis.

4.      Strengthened the discussion on the relationship between VR sensory evaluation and participants' emotional states, noting an increase in positive effects and a decrease in negative effects observed in the results.

Acknowledged the need for further research to comprehensively explore the relationship between PANAS and VR sensory analysis, particularly in understanding emotional responses and implications for sensory assessment in virtual environments

Comments on the Quality of English Language

Needed for improvement 

Author Response

Thank you for your kind work. Attached please find our answers to your questions. 

Reviewer 2 Report

Comments and Suggestions for Authors

Dear authors, in my opinion, this work is definitely of interest to the reader of the journal. The text is very clear.

Author Response

(The authors gave the same response as above.)

Reviewer 3 Report

Comments and Suggestions for Authors

Dear Author

the paper is quite interesting investigating the relationship between PANAS and VR sensory analysis.

Only minor revisions are required, but a deep revision of the English language. Sometimes is difficult to understand the meaning of the text.

In all the text, please verify the use of the abbreviation VR and the use of Virtual Reality (VR). After the first time you explained the meaning of VR (line 26), it is not necessary to use Virtual Reality (VR).

You can use VR or alternatively Virtual Reality (not boths)

For example in line 38 you can use Virtual Reality and eliminate VR

Verify the title 1.3: you can use only VR like in titles 1.2 and 1.5, the same in the capture of Table 1, line 421.The same in the Title 3.3: Post-VR questionnaire and in the capture of Figure 9.

The title 3.2 is missing.

Other suggestions

Line 15: add the meaning of the abbreviation of PANAS

Line 54: Moreover, sensory analysis refers to a method used to evaluate and describe ...Do you mean this:

Moreover, sensory analysis allows to evaluate and describe ....

Line 98-99: Additionally, the use of VR in history education has been found to promote...maybe:

Additionally, it was found that the use of VR in history education can promote....

Line 250-251: One student assistant was had been recruited to help set up the system and instruct the participants on the rules to follow during the experiment (or on the rules of the experiment)

Line 262: do you mean this:

While in the The virtual sensory booths these  had the size of 1m x 1m x 2.5m (w x d x h) and were completed with a  computer, a monitor, a chair, and samples indicated indicator with a three (3) digit randomized code  and a glass of water,.

Line 279-280 modify the sentence, do you mean this:

First, it was necessary to fill out online forms with demographic questions, using a tablet.

Line 281-292: please revise and improve all this paragraph, it is difficult to understand.

Line 302-303: ..please clarify the meaning of  .and task from the following measures.?

Line 362-367: please imporve the description of the results, and add the significative differences in Figure 4.

Line 371-373: improve the sentence:

Previous studies showed that VR was associated with positive emotion increases and negative emotion decreases.

Line 373 Revise the citation Yeo et al.19

Line 405-408: Improve this sentence, this is only a suggestion

 There are several emotions with a significtaive that had a significance before and after the experiment. The positive emotions that had significance differences and increased after the experiment are "Interested", "Excited", "Proud", and "Inspired",  which the emotions increase. Wwhile the Negative emotions which decreased are "Guilty", "Ashamed", "Nervous", and "Afraid" (Table 3).which the emotions decrease.

Lines 446-453: revise the style of the character, you used the same style of the captions of the figures 6 and 7.

Line 447-448 please improve this sentence, do you mean this:

This can be validated further from by Figure 4 that showing that the positive emotions increase while negative emotions decrease after the experiment.

Line 456: It was also explored the role of VR in eliciting positive emotions has also been explored in various contexts.

Line 463-465: please, modify the sentence, its is not clear, do you mean this?:

Furthermore, it was investigated the impact of VR on emotional empathy has been investigated, with reporting that VR increases emotional empathy. In particular, Martingano et al. [50] suggested ing the potential of VR to enhance positive emotional connections.

Line 465-468: please, modify the sentence, it is not clear, do you mean this:

Additionally, it was shown that the use of VR in mindfulness skills training exercises has been shown to reduces negative emotions and increases positive emotions in individuals. Gomez et al. [51] evidenced indicating the potential of VR in promoting positive emotional well-being.

Lines 477-487, 504-510 and 517-525: justify these paragraphs.

VLine 511-514: please, revise this sentence, it is not clear:

Various studies exploring the impact of VR on emotional experiences. conducted a review of research in VR, focusing on the perception of fear cues, emotion, and presence, aiming to identify the most relevant aspects of emotional experience...

line 518-519 Please improve this sentence

It was shown that VR has been show to influenceaffective states, ...

Line 519: Morover, it was found that VR  has also been found can enhance sensory ....

Comments on the Quality of English Language

Dear Authors,

please check the use of the  tenses of the verbs in the text.

Line 26, 32, 38, 50, 72, 101, 146, 152, 325: remove has or have before the verb

Line 107: maybe in a specific ..(put the article before specific environment)

Line 121: maybe ....that VR has a significant potential ..(put the article before significant potential)

Line 127: has created maybe creates

Line 128:.....and it has the capacity of....

Line 131: VR has a huge potential

Line 147: The PANAS has been is translated...and has been validated in...

Line 152:-153 The PANAS has demonstrated a good internal ...and a convergent,,,

Line 168: ,,,,,affective states and has been widely used in research and clinical settings.

Line 187, 195, 201 :,,,,has been was used ...

Line 198: of, coffee: eliminate the comma after coffee

Line 256:....had been was developed and designed using...

Line 262: While in the The virtual sensory booths, these are were completed...

Revise Line 495: maybe was used instead of is used ?

Author Response

Thank you for your detailed work. Attached please find our answers to your questions. 
